# Trajectory tracking sliding mode control for vertical take-off and landing aircraft based on double loop and global Lipschitz stability

**Liang Du** * 

Wuxi Vocational College of Science and Technology, Wuxi, Jiangsu, China

\* kingxm_edu@126.com

## Abstract

Vertical Take-Off and Landing (VTOL) aircraft excel in their ability to maneuver in limited spaces, making them ideal for a variety of uses including urban air mobility, emergency response, and disaster surveillance. Their agility and quick deployment features are especially valuable for executing complex missions in challenging environments. This paper addresses this issue by proposing a dual-loop sliding mode control (SMC) strategy optimized for VTOL models. However, tracking errors in the inner loop can impact the performance of the outer loop, complicating the assessment of the inner loop's convergence speed to meet the outer loop's criteria, and thus hindering the achievement of absolute stability in both control loops. To tackle this issue, the paper leverages the global asymptotic stability theorem for dynamic systems and develops a closed-loop system with global Lipschitz continuity, guaranteeing robust stability across both loops. This method not only bolsters the system's dependability but also enhances its flexibility to operate effectively under complex dynamic conditions, thereby increasing the overall resilience and performance of the VTOL control systems. The implementation of the sliding mode control strategy in VTOL models significantly enhances operational stability and reduces tracking errors in complex environments. Numerical simulations demonstrate that our approach reliably improves both performance and adaptability of the system under varying dynamic conditions.

**Citation:** Du L (2025) Trajectory tracking sliding mode control for vertical take-off and landing aircraft based on double loop and global Lipschitz stability. PLoS ONE 20(2): e0318294. https://doi.org/10.1371/journal.pone.0318294

**Data Availability Statement:** All relevant data are within the manuscript and its Supporting information files.

**Funding:** This research is funded by the General Project of Philosophy and Social Sciences

## 1 Introduction

The advent of advanced aerial vehicles has underscored a critical need for sophisticated control systems capable of managing their operation in diverse environments. Particularly, the inherent nonlinear nature of these systems, coupled with complex dynamic interactions, presents a formidable challenge in achieving stable and efficient flight [1]. This complexity is primarily due to the intricate relationships between various aerodynamic forces and control inputs, which often result in highly interdependent behavioral characteristics (e.g., roll, pitch, and yaw).

Research in Jiangsu Province Higher Education Institutions (Project Title: "Research on Entrepreneurial Models for College Students in Higher Vocational Colleges Based on 'Co-Creation between Teachers and Students," Project No. 2024SJYB0726).

**Competing interests:** The authors have declared that no competing interests exist.

Vertical take-off and landing (VTOL) aircraft are capable of ascending and descending vertically, which makes them well-suited for use in tight spaces like crowded cities or remote locations [2–5]. These aircraft play a significant role in various fields including urban air mobility, emergency rescue, and logistical support. VTOL aircraft are pivotal in sectors such as urban air mobility and emergency rescue, thanks to their ability to operate effectively in tight spaces. This capability allows VTOL aircraft to be rapidly deployed to hard-to-reach areas, making them ideal for transportation in densely populated urban areas and remote locations. As emphasized in the research by Bacchini and Cestino [6], this rapid deployment capability of VTOLs is crucial for carrying out diverse tasks in complex and demanding environments, significantly enhancing the efficiency of emergency response and logistical operations. Considering the expansion of urban areas and the growing demand for efficient transportation solutions, VTOL technology is expected to play a key role in the future mobility systems, as highlighted by Orbea's research [7]. However, the current VTOL control system also faces some challenges. Such as the integration of autonomous operation in urban environment, the management of airspace congestion and the need to improve the robustness to environmental interference.

The unique design features of VTOL aircraft have inspired significant research into their various models and possible uses. In the research conducted by Abd et al [8], the control system of an octocopter is upgraded with a new structure that boosts aerodynamic efficiency and demonstrates fault tolerance in rotor failures during experimental flights. In the study by Dalamagkidis [9], an enhanced unmanned ground vehicle was engineered to facilitate the transport of VTOL aircraft to designated areas, significantly broadening their operational scope. This approach not only enhances the functional capabilities of the aircraft but also increases the variety of missions they can undertake. Meanwhile, Ucgun introduced a cutting-edge preflight testing system for VTOL aircrafts, featuring a three-dimensional nested circle testing platform, a user-friendly graphical interface, and a wireless communication setup. This system allows for thorough axis testing prior to flights, enhancing safety by reducing the risk of inflight failures [10]. In [11], which aim is to establish a mathematical model of the dynamics for VTOL aircrafts, which will support the development of control laws for inherently unstable multirotor aircraft. In addition, this model will be utilized to design algorithms for accurately estimating the orientation of target objects. In [12], the researchers conducted an in-depth systematic classification of various types of VTOL aircraft and compared the performance metrics, such as range and flight time, of several representative designs. Additionally, the article explores the potential applications of VTOL aircraft in precision agriculture and proposes future research directions and development recommendations. These analyses lay an important foundation for enhancing the performance of new VTOL aircraft and further research, aiming to promote innovation and practical applications of related technologies, particularly in improving efficiency in fields such as agriculture. Through this research, the document provides valuable data references for the academic community and practical guidance for industry practices. The study reported in [13] developed a modular VTOL drone designed for flexible assembly and versatile operation to test distributed estimation and control algorithms. Through the adjustment of control parameters, this drone showcased enhanced adaptability and performance in different configurations.

In contrast to the aforementioned research, many researchers have done a lot of research on the trajectory tracking control method of VTOL aircraft [14–16]. Lu and Yan, along with their colleagues, have proposed a particle swarm optimization algorithm enhanced by PID control for tackling parameter identification in VTOL aircraft. The experimental results demonstrate that this particular version of the particle swarm optimization algorithm outperforms others in solving a majority of multimodal optimization challenges, due to its improved search

capabilities and efficiency. However, this method does not take into consideration the external disturbances that VTOL aircraft are likely to face [17]. In Reference [18], a nonlinear optimal control method is proposed, along with the design of an $H_\infty$ feedback controller, which ensures the rapid and precise tracking of all state variables under model uncertainty and external disturbances. The stability of the control scheme is confirmed using Lyapunov analysis. In [19], Haq, Izhar Ul and his team developed a new strategy that combines a nonlinear generalized global sliding mode controller with a feedforward neural network, using a DC-DC converter to efficiently extract maximum power from photovoltaic arrays. Simulation results demonstrated that this method exhibits outstanding efficiency and rapid response capabilities under varying environmental conditions. In [20], the researchers propose a unified controller based on incremental nonlinear dynamic inversion, specifically designed for VTOL transition aircraft. The aim is to integrate various flight phases into a single universal controller. This controller effectively addresses the non-affine characteristics of control inputs, thereby simplifying the flight control process and allowing for operation with just one baseline controller throughout the entire flight envelope. Zou and Meng introduced an innovative distributed estimation approach designed to address the cooperative trajectory tracking problem among several VTOL aircraft. Utilizing a hierarchical system design and an auxiliary mechanism, their method effectively manages the underactuation characteristics of the aircraft, ensuring efficient synchronization of their trajectories [21]. Building on this concept, Wang and his team addressed similar tracking issues for VTOL aircrafts through an interference suppression control strategy that integrates a nonlinear error model with hierarchical control, thereby ensuring effective management and overall stability [22]. Huang proposed a position-based distributed formation control for VTOL aircraft swarms that operate under unknown disturbances, incorporating adaptive control laws and auxiliary systems to enhance the formation's efficacy and stability, despite external disruptions [15]. Lin's research introduced a transition control strategy for VTOL aircrafts facing severe environmental disturbances, utilizing a dynamic model and an adaptive sliding mode observer to maintain control effectiveness and stability [23]. Wen, Chen, and colleagues investigated the application of a discrete active disturbance rejection controller in tilt-rotor UAV designs, leveraging a dynamic model to achieve enhanced tracking accuracy and stability in their simulations [24].

In our research on VTOL aircraft and their control systems, we emphasize the following key contributions: We employed sliding mode control techniques, known for their robustness against disturbances, to improve the operational stability and responsiveness of VTOL systems.

1. We have engineered an innovative sliding mode control strategy that employs a dual-loop structure, effectively reducing the influence of adjustments within the inner loop on the performance of the outer loop. This approach greatly enhances the response rate and stability of VTOL aircraft, thereby increasing their proficiency in managing dynamic environmental conditions.

2. We utilized the global asymptotic stability theorem to devise a closed-loop system that exhibits global Lipschitz continuity. This arrangement ensures robust stability throughout both the inner and outer loops of the control system. Such a theoretical improvement greatly enhances the reliability of VTOL aircraft in dynamic environments, offering a strong base for practical implementations in real-world situations.

3. Through rigorous numerical simulations conducted in MATLAB/SIMULINK, we have comprehensively tested and validated our proposed control strategy. These simulations serve to affirm the feasibility of our design and establish a robust theoretical foundation,

paving the way for future advancements and practical applications in the field of VTOL aircraft technology.

The remaining structure of this paper is as follows: Section 2 provides a detailed description of the problem and the transformation of the VTOL model, along with relevant lemmas and assumptions. Section 3 first clarifies the control objectives of this study, then designs a robust sliding mode controller based on a dual-loop structure, and analyzes its stability. In Section 4, the efficacy of the proposed control strategy is validated using MATLAB simulations. Section 5 concludes the paper with a summary and provides a brief overview of possible avenues for future research.

## 2 Problem description and model transformation

### 2.1 VTOL model description

**Remark 1**: R.O. Saber has devised a specialized decoupling algorithm for underactuated systems, enhancing control strategies by simplifying the complex coupled dynamic behaviors into manageable subsystems [25, 26]. This algorithm constructs a mathematical model, applies transformations to achieve decoupling, and facilitates the use of standard controllers on decomposed subsystems, thereby enhancing system resilience to disturbances and uncertainties [27]

Using the mechanism analysis method [28, 29], we can derive the dynamic equilibrium equations for VTOL aircraft

$$\begin{cases} \ddot{x} = -u_1 \sin\theta + \varepsilon u_2 \cos\theta \\ \ddot{y} = u_1 \cos\theta + \varepsilon u_2 \sin\theta - g \\ \ddot{\theta} = u_2 \end{cases} \tag{1}$$

where, $u_1$ and $u_2$ are the control inputs, with $u_1$ corresponding to the thrust moment produced at the aircraft's base and $u_2$ associated with the roll moment. Furthermore, $g$ stands for the gravitational acceleration, and $\varepsilon$ represents a coefficient that defines the interaction between $u_1$ and $u_2$. These elements are crucial in forming the aircraft's dynamic model, which is essential for both comprehending and managing its flight dynamics.

By defining $x_1 = x, x_2 = \dot{x}, y_1 = y, y_2 = \dot{y}, \theta_1 = \theta$, and $\theta_2 = \omega$, the model Eq (1) can be rewritten as follows

$$\begin{cases} \dot{x}_1 = x_2 \\ \dot{x}_2 = -u_1 \sin\theta + \varepsilon u_2 \cos\theta \\ \dot{y}_1 = y_2 \\ \dot{y}_2 = u_1 \cos\theta + \varepsilon u_2 \sin\theta - g \\ \dot{\theta} = \omega \\ \dot{\omega} = u_2 \end{cases} \tag{2}$$

Drawing on the decoupling algorithm introduced by R.O. Saber [29], and for Eq (2), taking $q_1 = x_1, p_1 = x_2, q_2 = \theta, p_2 = \omega$, we have $g_1(q_2) = \varepsilon \cos\theta$ and $g_2(q_2) = 1$. Thus, $\frac{g_1(\theta)}{g_2(\theta)} = \varepsilon \cos\theta$, and

$\int_0^0 \frac{g_1(s)}{g_2(s)} ds = \int_0^0 \varepsilon \cos \theta d\theta = \varepsilon \sin \theta$. The corresponding decoupling algorithm is as follows:

$$
\begin{cases}
z_1 = x_1 - \varepsilon \sin \theta \\
z_2 = x_2 - \varepsilon \cos \theta \cdot \omega \\
\xi_1 = \theta \\
\xi_2 = \omega
\end{cases} \tag{3}
$$

For Eq (2), taking $q_1 = y_1$, $p_1 = y_2$, $q_2 = \theta$, $p_2 = \omega$, we have $g_1(q_2) = \varepsilon \sin \theta$ and $g_2(q_2) = 1$. Thus, $\frac{g_1(\theta)}{g_2(\theta)} = \varepsilon \sin \theta$, and $\int_0^\theta \frac{g_1(s)}{g_2(s)} ds = \int_0^\theta \varepsilon \sin s ds = -\varepsilon \int_0^\theta (\cos s)' ds = -\varepsilon (\cos \theta - 1)$. The corresponding decoupling algorithm is as follows:

$$
\begin{cases}
w_1 = y_1 + \varepsilon(\cos \theta - 1) \\
w_2 = y_2 - \varepsilon \sin \theta \cdot \omega \\
\xi_1 = \theta \\
\xi_2 = \omega
\end{cases} \tag{4}
$$

Given the equations:

$$
\begin{aligned}
\dot{z}_2 &= \dot{x}_2 + \varepsilon \sin \theta \cdot \omega^2 - \varepsilon \cos \theta \cdot \dot{\omega} = -u_1 \sin \theta + \varepsilon \cos \theta \cdot u_2 + \varepsilon \sin \theta \cdot \omega^2 - \varepsilon \cos \theta \cdot u_2 \\
&= -u_1 \sin \theta + \varepsilon \sin \theta \cdot \omega^2 = -\bar{u}_1 \sin \theta
\end{aligned} \tag{5}
$$

$$
\begin{aligned}
\dot{w}_2 &= \dot{y}_2 - \varepsilon \cos \theta \cdot \omega^2 - \varepsilon \sin \theta \cdot \dot{\omega} = u_1 \cos \theta + \varepsilon u_2 \sin \theta - g - \varepsilon \cos \theta \cdot \omega^2 - \varepsilon \sin \theta \cdot u_2 \\
&= u_1 \cos \theta - g - \varepsilon \cos \theta \cdot \omega^2 = \bar{u}_1 \cos \theta - g
\end{aligned} \tag{6}
$$

where $\bar{u}_1 = u_1 - \varepsilon \omega^2$.

Then, the corresponding decoupling algorithm can be derived

$$
\begin{cases}
z_1 = x_1 - \varepsilon \sin \theta \\
z_2 = x_2 - \varepsilon \cos \theta \cdot \omega \\
w_1 = y_1 + \varepsilon(\cos \theta - 1) \\
w_2 = y_2 - \varepsilon \sin \theta \cdot \omega \\
\xi_1 = \theta \\
\xi_2 = \omega
\end{cases} \tag{7}
$$

After decoupling, the model Eq (2) becomes:

$$
\begin{cases}
\dot{z}_1 = z_2 \\
\dot{z}_2 = -\bar{u}_1 \sin \theta \\
\dot{w}_1 = w_2 \\
\dot{w}_2 = \bar{u}_1 \cos \theta - g
\end{cases} \tag{8}
$$

where, $\bar{u}_1 = u_1 - \varepsilon \omega^2$.

Set the following variables:

$$\begin{cases} v_1 = -\bar{u}_1 \sin \theta \\ v_2 = \bar{u}_1 \cos \theta - g \end{cases} \tag{9}$$

Then the model can be transformed into:

$$\begin{cases} \dot{z}_1 = z_2 \\ \dot{z}_2 = v_1 \\ \dot{w}_1 = w_2 \\ \dot{w}_2 = v_2 \\ \dot{\theta} = \omega \\ \dot{\omega} = u_2 \end{cases} \tag{10}$$

**Remark 2**: Employing a decoupling and model transformation strategy for VTOL systems, as opposed to traditional direct design methods for UAV trajectory controllers, offers significant advantages [30–33]. This strategy simplifies the aircraft's dynamics into independent subsystems, improving control accuracy, system responsiveness, and stability, thus ensuring robust trajectory tracking under diverse conditions [34].

## 2.2 Useful lemmas and assumptions

**Assumption 1**: It is assumed that the VTOL aircraft's dynamic behavior can be effectively modeled as a linear system within a narrowly defined operational range, which substantially simplifies the design of the controller.

**Assumption 2**: It is assumed that all necessary system states—like position, velocity, and attitude—are precisely measurable or estimable through advanced observers, we ensure that the control algorithm remains robust and effective.

**Lemma 1** [35]: The following global asymptotic stability theorem is proposed, applicable to the following dynamic system:

$$\begin{aligned} \dot{\gamma}_1 &= \gamma_2 \\ \dot{\gamma}_2 &= -\alpha \tanh \left( k\gamma_1 + l\gamma_2 \right) - \beta \tanh \left( l\gamma_2 \right) \end{aligned} \tag{11}$$

where, $\alpha, \beta, k, l > 0$. According to this theorem, as time $t \to \infty$, the system states $\gamma_1$ and $\gamma_2$ will converge to zero, indicating that the system exhibits global asymptotic stability.

**Proof 1**: According to reference [35], we consider the function $\cosh(x) = \frac{e^{-x} + e^x}{2}$, which is always greater than or equal to 1; thus, $\ln(\cosh(x))$ is also non-negative. When $x = 0$, the value of $\ln(\cosh(x))$ is 0. To prove that as $t \to \infty$, the variables $\gamma_1$ and $\gamma_2$ tend to 0, we define a Lyapunov function

$$V = \alpha \ln \left( \cosh \left( k\gamma_1 + l\gamma_2 \right) \right) + \beta \ln \left( \cosh \left( l\gamma_2 \right) \right) + \frac{1}{2} k\gamma_2^2 \tag{12}$$

Then

$$
\begin{aligned}
\dot{V} &= \alpha \frac{\sinh(k\gamma_1 + l\gamma_2)}{\cosh(k\gamma_1 + l\gamma_2)} (k\dot{\gamma}_1 + l\dot{\gamma}_2) + \beta \frac{\sinh(l\gamma_2)}{\cosh(l\gamma_2)} l\dot{\gamma}_2 + k\gamma_2\dot{\gamma}_2 \\
&= \alpha(k\dot{\gamma}_1 + l\dot{\gamma}_2)\tanh(k\gamma_1 + l\gamma_2) + \beta l\dot{\gamma}_2 \tanh(l\gamma_2) + k\gamma_2\dot{\gamma}_2
\end{aligned}
\tag{13}
$$

Let $t_1 = \tanh(k\gamma_1 + l\gamma_2)$ and $t_2 = \tanh(l\gamma_2)$, then we have $\dot{\gamma}_2 = -\alpha t_1 - \beta t_2$. Substituting this expression, we can write $\dot{V}$ as follows:

$$
\begin{aligned}
\dot{V} &= \alpha(k\gamma_2 + l(-\alpha t_1 - \beta t_2))t_1 + \beta l(-\alpha t_1 - \beta t_2)t_2 + k\gamma_2(-\alpha t_1 - \beta t_2) \\
&= \alpha k\gamma_2 t_1 - l\alpha^2 t_1^2 - l\alpha\beta t_2 t_1 - l\beta\alpha t_1 t_2 - l\beta^2 t_2^2 - k\alpha\gamma_2 t_1 - k\beta\gamma_2 t_2 \\
&= -l(\alpha^2 t_1^2 + 2l\alpha\beta t_2 t_1 + \beta^2 t_2^2) - k\beta\gamma_2 t_2 \\
&= -l(\alpha t_1 + \beta t_2)^2 - k\beta\gamma_2 t_2
\end{aligned}
\tag{14}
$$

Since $x \tanh(x) = x\frac{e^x - e^{-x}}{e^x + e^{-x}} \geq 0$, it follows that $\gamma_2 t_2 = \gamma_2 \tanh(l\gamma_2) \geq 0$. Thus, we have $\dot{V} \leq 0$, and this holds true if and only if the following conditions are satisfied.

When $\gamma_1 = \gamma_2 = 0$, we have $\dot{V} = 0$. As $t \to \infty$, both $\gamma_1 \to 0$ and $\gamma_2 \to 0$, and the rate of convergence of the system depends on $\alpha, \beta, k$, and $l$. Furthermore, since $\dot{\gamma}_2 = -\alpha \tanh(k\gamma_1 + l\gamma_2) - \beta \tanh(l\gamma_2)$, as $\gamma_1 \to 0$ and $\gamma_2 \to 0$, it follows that $\dot{\gamma}_2 \to 0$.

Since $\tanh(x) = \frac{e^x - e^{-x}}{e^x + e^{-x}}$ takes values in the range of $[-1, 1]$, we can derive the following inequality:

$$
|\dot{\gamma}_2| = |-\alpha \tanh(k\gamma_1 + l\gamma_2) - \beta \tanh(l\gamma_2)| \leqslant \alpha + \beta
\tag{15}
$$

Thus, if we design the control law according to the structure of the model (11), it ensures that the control input remains bounded, with an upper bound of $\alpha + \beta$. This means that through appropriate design, we can limit the fluctuations of the control input, allowing the system to operate within a controllable range, thereby enhancing the stability and reliability of the system.

**Lemma 2**: Lipschitz continuity, a fundamental property in mathematical analysis and control theory, involves a condition where there exists a constant $L$ such that for any two points $x$ and $y$ in a domain, the inequality:

$$
|f(x) - f(y)| \leq L|x - y|
\tag{16}
$$

holds, where $L$ is known as the Lipschitz constant. This property is crucial for ensuring that small changes in the input of a function lead to proportionally small changes in the output, which is vital for the stability and robustness of control systems.

**Assumption 3**: It is assumed that our model assumes an accurate representation of the VTOL aircraft's dynamics and asserts that the decoupling of these dynamics can be successfully executed, reflecting true aircraft behavior.

## 3 Robust sliding mode controller design

### 3.1 Control objective

The control objective of this paper is to enhance the operational stability and responsiveness of VTOL aircraft in complex environments by designing a system that employs a dual-loop sliding mode control strategy. This control approach minimizes the impact of adjustments within the inner loop on the performance of the outer loop and optimizes the convergence speed of

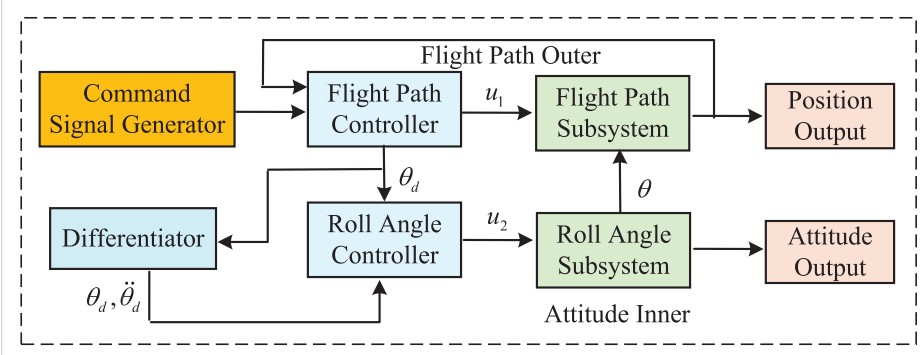

**Fig 1. The control logic structure diagram.**

the outer loop. By utilizing the global asymptotic stability theorem, the system ensures robust stability under various dynamic conditions. The control logic diagram, which will be displayed, illustrates the VTOL model, dual-loop structure, control signal pathways, and stability analysis, clearly articulating how these control objectives are achieved. The control logic diagram is shown in Fig 1.

## 3.2 Design of tracking control algorithm

For the first subsystem in Eq (1), which is the trajectory tracking subsystem, it serves as the outer loop, with the control input being $u_1$. The state equations for this subsystem are:

$$\begin{cases} \dot{z}_1 = z_2 \\ \dot{z}_2 = v_1 \\ \dot{w}_1 = w_2 \\ \dot{w}_2 = v_2 \end{cases} \tag{17}$$

Control laws are designed for the $z$ and $w$ subsystems in Eq (17). First, for the $z$ subsystem, let the reference value of $z_1$ be $z_{1d}$, and define the errors as $\tilde{z}_1 = z_1 - z_{1d}$ and $\dot{\tilde{z}}_1 = z_2 - \dot{z}_{1d}$. The model becomes:

$$\begin{cases} \dot{\tilde{z}}_1 = \tilde{z}_2 \\ \dot{\tilde{z}}_2 = v_1 - \ddot{z}_{1d} \end{cases} \tag{18}$$

According to Lemma 1, if the control law is designed as:

$$v_1 = -\alpha_1 \tanh\left(k_1 \tilde{z}_1 + l_1 \tilde{z}_2\right) - \beta_1 \tanh\left(l_1 \tilde{z}_2\right) + \ddot{z}_{1d} \tag{19}$$

then $\tilde{z}_1 \to 0$ and $\dot{\tilde{z}}_1 \to 0$ can be achieved, where $\alpha_1, \beta_1, k_1, l_1 > 0$.

Similarly, for the $w$ subsystem, let the reference value of $w_1$ be $w_{1d}$, and define the errors as $\tilde{w}_1 = w_1 - w_{1d}$ and $\dot{\tilde{w}}_1 = w_2 - \dot{w}_{1d}$. The model becomes:

$$\begin{cases} \dot{\tilde{w}}_1 = \tilde{w}_2 \\ \dot{\tilde{w}}_2 = v_2 - \ddot{w}_{1d} \end{cases} \tag{20}$$

Again, according to Lemma 1 and Lemma 2, if the control law is designed as:

$$v_2 = -\alpha_2 \tanh\left(k_2\tilde{w}_1 + l_2\tilde{w}_2\right) - \beta_2 \tanh\left(l_2\tilde{w}_2\right) + \ddot{w}_{1d} \tag{21}$$

By applying Lemma 1, designing bounded control inputs $v_1$ and $v_2$, the closed-loop systems (18) and (20) can be ensured to be globally Lipschitz continuous, thus ensuring the boundedness of $z_1$ and $w_1$.

Given that the control inputs are $v_1 = -\bar{u}_1 \sin\theta$ and $v_2 = \bar{u}_1 \cos\theta - g$, and considering that the direction is upwards relative to the direction of gravity, meaning that $u_1$ is always in the positive direction, the control law and the angle required to achieve it are derived as follows:

$$\begin{cases} \bar{u}_1 = \sqrt{v_1^2 + (v_2 + g)^2} \\ \theta_d = \arctan\left(-\dfrac{v_1}{v_2 + g}\right) \end{cases} \tag{22}$$

**Remark 3**: To avoid $v_2 + g$ being zero, the condition $|v_2| < g$ must be satisfied. According to the control law in Eq (21), we have $|v_2| \leq \alpha_2 + \beta_2 + \max \ddot{w}_d$, so the design of $\alpha_2$ and $\beta_2$ should satisfy $\alpha_2 + \beta_2 + \max\{\ddot{w}_{1d}\} < g$. In a VTOL aircraft, the value and variation range of the angle $\theta$ are small, so the value and variation range of the angle command $\theta_d$ are also small. It is reasonable to set the angle command $\theta_d$ in the range of $-\frac{\pi}{3} \leq \theta_d \leq \frac{\pi}{3}$. Since $\theta_d = \arctan\left(-\frac{v_1}{v_2+g}\right)$, the condition $-\frac{\sqrt{3}}{2} \leq -\frac{v_1}{v_2+g} \leq \frac{\sqrt{3}}{2}$ must hold, which implies $\left|\frac{v_1}{v_2+g}\right| \leq \frac{\sqrt{3}}{2}$. Hence:

$$|v_1| \leq \frac{\sqrt{3}}{2}|v_2 + g| \tag{23}$$

The values of $\alpha_1$ and $\beta_1$ must satisfy the above condition, that is:

$$\max\left| -\alpha_1 \tanh\left(k_1\tilde{z}_1 + l_1\tilde{z}_2\right) - \beta_1 \tanh\left(l_1\tilde{z}_2\right) + \ddot{z}_{1d}\right|$$
$$\leq \frac{\sqrt{3}}{2}\min\left| -\alpha_2 \tanh\left(k_2\tilde{w}_1 + l_2\tilde{w}_2\right) - \beta_2 \tanh\left(l_2\tilde{w}_2\right) + \ddot{w}_{1d} + g\right| \tag{24}$$

By $u_1 = u_1 - \varepsilon\omega^2$, the actual control input becomes:

$$u_1 = \bar{u}_1 + \varepsilon\omega^2 \tag{25}$$

In the controller depicted above, the outer-loop system generates the desired angle command $\theta_d$, which is forwarded to the inner-loop system. The difference $\theta - \theta_d$, representing the error in the outer loop, is corrected by the control mechanisms in the inner loop.

In the discussion above, the second subsystem is a roll tracking system, namely the inner loop, with its control input $u_2$, designed to achieve precise tracking of the outer loop angle $\theta_d$.

The dynamics are described by:

$$\begin{cases} \dot{\theta} = \omega \\ \dot{\omega} = u_2 \end{cases} \tag{26}$$

We define the angle command $\theta_d$, with tracking error $\eta = \theta - \theta_d$. A sliding mode function is introduced, $s = c\eta + \dot{\eta}$, where $c > 0$. The Lyapunov function is defined as $V_3 = \frac{1}{2}s^2$. Thus,

$$\dot{s} = c\dot{e} + \ddot{e} = c(\dot{\theta} - \dot{\theta}_d) + \ddot{\theta} - \ddot{\theta}_d = c(\omega - \dot{\theta}_d) + u_2 + \Delta_s(t) - \ddot{\theta}_d \tag{27}$$

In the context of our study, the SMC law is designed to ensure that the state trajectory of the VTOL aircraft reaches and maintains a sliding condition defined by the sliding surface $s = c\eta + \dot{\eta}$, where $\eta = \theta - \theta_d$ is the angle tracking error. The control law is formulated as:

$$u_2 = -c(\omega - \dot{\theta}_d) + \ddot{\theta}_a - ks \tag{28}$$

where $k > 0$. This results in $\dot{s} = -ks$, leading to $\dot{V}_3 = -ks^2 = -2kV_3$, hence $V_3(t) = e^{-3k(1-t_0)}V_3(t_0)$. The control system exhibits exponential convergence, with the rate of convergence dependent on the parameter $k$. By adjusting the value of $k$, we can ensure a rapid response of $\theta$ to $\theta_d$.

**Remark 4**: A third-order integrator chain differentiator is used to compute the derivatives of signals, enabling precise tracking and dynamic adjustment of target signals in control systems [36–38]. In control rule (28), since directly differentiating $\theta_d$ from Eq (22) is quite complex, we can utilize a third-order integrator chain differentiator to simplify the calculation. This differentiator uses the following equations to compute $\dot{\theta}_d$ and $\ddot{\theta}_d$:

$$\begin{cases} \dot{x}_1 = x_2 \\ \dot{x}_2 = x_3 \\ \dot{x}_3 = -\dfrac{k_1}{\varepsilon^3}(x_1 - \theta_d) - \dfrac{k_2}{\varepsilon^2}x_2 - \dfrac{k_3}{\varepsilon}x_3 \end{cases} \tag{29}$$

To avoid peak phenomena during the initial operation of the differentiator, we set $\varepsilon = \frac{1}{100}(1 - e^{-2t})$ for the interval $0 \leqslant t \leqslant 1.0$. This setting helps suppress initial peak errors without sacrificing dynamic response speed.

## 3.3 Stability analysis of closed-loop system

The stability of the aforementioned closed-loop system is achieved under the premise that the attitude angle $\theta$ quickly and accurately tracks $\theta_d$. Under ideal conditions, the control laws are described as:

$$\begin{cases} v_1 = -\bar{u}_1 \sin\theta_d \\ v_2 = \bar{u}_1 \cos\theta_d - g \end{cases} \tag{30}$$

If there is a discrepancy between the actual angle $\theta$ and the desired angle $\theta_d$, it will inevitably affect the stability of the position closed-loop system. If the influence of angular tracking error is considered, using the control laws $v_1$ and $v_2$ under ideal conditions, the expressions for the

closed-loop system can be developed based on Eqs (18) and (20).

$$\begin{cases} \dot{\bar{z}}_1 = \bar{z}_2 \\ \dot{\bar{z}}_2 = -\bar{u}_1 \sin\theta - \ddot{z}_{1d} = v_1 - \ddot{z}_{1d} + \bar{u}_1(\sin\theta_d - \sin\theta) \end{cases} \tag{31}$$

$$\begin{cases} \dot{\tilde{w}}_1 = \tilde{w}_2 \\ \dot{\tilde{w}}_2 = \bar{u}_1 \cos\theta - g - \ddot{w}_{1d} = v_2 - \ddot{w}_{1d} + \bar{u}_1(\cos\theta - \cos\theta_d) \end{cases} \tag{32}$$

After substituting Eqs (19) and (21), we obtain the following system of equations:

$$\begin{cases} \dot{\tilde{z}}_1 = \tilde{z}_2 \\ \dot{\tilde{z}}_2 = -\alpha_1 \tanh(k_1\bar{z}_1 + \iota_1\bar{z}_2) - \beta_1 \tanh(l_1\bar{z}_2) + \bar{u}_1(\sin\theta_d - \sin\theta) \end{cases} \tag{33}$$

$$\begin{cases} \dot{w}_1 = \tilde{w}_2 \\ \dot{\tilde{w}}_2 = -\alpha_2 \tanh(k_2\tilde{w}_1 + l_2\tilde{w}_2) - \beta_2 \tanh(l_2\tilde{w}_2) + \bar{u}_1(\cos\theta - \cos\theta_d) \end{cases} \tag{34}$$

We analyze the stability of $\tilde{z}_1$ to assess the performance of the closed-loop system. This is ensured by using $\ln(\cosh(x)) \geqslant 0$, where $\cosh(x) = \frac{e^{-x}+e^x}{2} \geqslant 1$, indicating the stability at $x = 0$.

According to the global Lipschitz condition, we ensure that $\bar{u}_1 = \sqrt{v_1^2 + (v_2 + g)^2}$ is bounded.

We consider the Lyapunov function:

$$V = \alpha_1 \ln(\cosh(k_1\tilde{z}_1 + l_1\tilde{z}_2)) + \beta_1 \ln(\cosh(l_1\tilde{z}_2)) + \frac{1}{2}k_1\tilde{z}_2^2 \tag{35}$$

where $\alpha_1, \beta_1, k_1, l_1 > 0$. Then, the time derivative of $V$ is:

$$\dot{V} = \alpha_1(k_1\dot{\tilde{z}}_1 + l_1\dot{\tilde{z}}_2)\tanh(k_1\tilde{z}_1 + l_1\tilde{z}_2) + \beta_1 l_1\dot{\tilde{z}}_2 \tanh(l_1\tilde{z}_2) + k_1\tilde{z}_2\dot{\tilde{z}}_2 \tag{36}$$

Define $t_1 = \tanh(k_1\tilde{z}_1 + l_1\tilde{z}_2)$, $t_2 = \tanh(l_1\tilde{z}_2)$, $t_3 = \bar{u}_1(\sin\theta d - \sin\theta)$, and $\dot{\tilde{z}}_2 = -\alpha_1 l_1 - \beta_1 t_2 + t_3$. With these definitions, we can rewrite $\dot{V}$ as follows:

$$\begin{aligned} \dot{V} &= \alpha_1(k_1\tilde{z}_2 + l_1(-\alpha_1 t_1 - \beta_1 t_2 + t_3))t_1 + \beta_1 l_1(-\alpha_1 t_1 - \beta_1 t_2 + t_3)t_2 \\ &\quad + k_1\tilde{z}_2(-\alpha_1 t_1 - \beta_1 t_2 + t_3) \\ &= \alpha_1 k_1\tilde{z}_2 t_1 - l_1^2 t_1^2 - l_{\alpha_1}\beta_1 t_2 t_1 - l_1\beta_1\alpha_1 t_1 t_2 - l_1\beta_1^2 t_2^2 - k_1\alpha_1\tilde{z}_2\iota_1 \\ &\quad - k_1\beta_1\tilde{z}_2 t_2 + \alpha_1 l_1 t_3 t_1 + \beta_1 l_1 t_3 t_2 + k_1\tilde{z}_2 t_3 \\ &= -l_1(\alpha_1^2 t_1^2 + 2l_1\alpha_1\beta_1 t_2 t_1 + \beta_1^2 t_2^2) - k_1\beta_1\tilde{z}_2 t_2 + t_3(\alpha_1 l_1 t_1 + \beta_1 l_1 t_2 + k_1\tilde{z}_2) \\ &= -l_1(\alpha_1 t_1 + \beta_1 t_2)^2 - k\beta_1\tilde{z}_2 t_2 + t_3(\alpha_1 l_1 t_1 + \beta_1 l_1 t_2 + k_1\tilde{z}_2) \end{aligned} \tag{37}$$

where

$$t_3(\alpha_1 l_1 t_1 + \beta_1 l_1 t_2 + k_1 \tilde{z}_2)$$

$$= \bar{u}_1(\sin\theta_d - \sin\theta)(\alpha_1 l_1 \tanh{(k_1\tilde{z}_1 + l_1\tilde{z}_2)} + \beta_1 l_1 \tanh{(l_1\tilde{z}_2)} + k_1\tilde{z}_2)$$

$$\leqslant \bar{u}_1|\eta_1|(\alpha_1 l_1 + \beta_1 l_1 + k_1\tilde{z}_2)$$

Considering the sine function identity:

$$\sin{(A + B)} - \sin{(A - B)} = 2\cos A \sin B \tag{38}$$

Let $A = \frac{\theta_d + \theta}{2}$ and $B = \frac{\theta_d - \theta}{2}$. This derivation leads to:

$$\sin\theta_d - \sin\theta = 2\cos\frac{\theta_d + \theta}{2}\sin\frac{\theta_d - \theta}{2} \tag{39}$$

Thus,

$$|\sin\theta_d - \sin\theta| = 2\left|\cos\frac{\theta_d + \theta}{2}\sin\frac{\theta_d - \theta}{2}\right| \leq 2\left|\sin\frac{\theta_d - \theta}{2}\right| \tag{40}$$

Thus,

$$|\eta_1| = |\sin\theta_d - \sin\theta| \leq 2\left|\sin\frac{(\theta_d - \theta)}{2}\right| \leq |\eta| \tag{41}$$

If the angular error $\eta = \theta - \theta_d$ decays exponentially, then $\eta_1 = \sin\theta_d - \sin\theta$ will also decay exponentially.

Therefore,

$$\dot{V} \leq -l_1(\alpha_1 l_1 + \beta_1 t_2)^2 - k_1\beta_1\tilde{z}_2 l_2 + \bar{u}_1|\eta_1|(\alpha_1 l_1 + \beta_1 l_1 + k_1\tilde{z}_2) \tag{42}$$

Moreover, since $x\tanh(x) \geq 0$, it follows that $k_1\beta_1\tilde{z}_2 t_2 \geqslant 0$.

Similarly, applying the same reasoning to the closed-loop system Eq (34), it can be demonstrated that as time $t$ approaches infinity, the error variables $\tilde{w}_1$ and $\tilde{w}_2$ will converge to zero.

Based on Eq (19), if the desired trajectory is set as $z_{1d} = t$, then its second derivative $\ddot{z}_{1d} = 0$. Employing the control law $v_1$, it ensures that as time $t$ progresses towards infinity, the tracking errors $\tilde{z}_1$ and $\tilde{z}_2$ both approach zero, and consequently, the control input $v_1$ also approaches zero, meaning that the acceleration $\ddot{z}_1 = -\bar{u}_1\sin\theta$ approaches zero, leading to $\theta \to 0$. Furthermore, since the system states are $z_1 = x_1 - \varepsilon\sin\theta$, $w_1 = y_1 + \varepsilon(\cos\theta - 1)$, as $\theta$ approaches zero, the system states $x_1$ and $y_1$ will respectively converge to $z_1$ and $w_1$, thereby achieving precise trajectory tracking.

## 4 Simulation verification

### 4.1 Simulation comparison

To validate the effectiveness of the control method proposed in this paper, we compare it with the PID control method outlined in Reference [17]. The comparison involves assessing both the proposed and PID control methods under similar conditions, with specific parameters and settings referenced from Literature 1. This approach allows us to directly evaluate the performance differences and potential improvements offered by our proposed method over the established PID control, thereby highlighting its advantages in practical applications. Further details on the PID control approach and its parameters can be found in Reference [17].

## 4.2 Simulation parameter setting

In this simulation, the control object is described by Eq (1), with the gravitational acceleration set to $g = 9.8$ m/s$^2$. The initial conditions are defined as follows: $x_1(0) = 1$, $x_2(0) = 0$, $y_1(0) = 1.0$, $y_2(0) = 0$, $\theta(0) = 0$, $\omega(0) = 0$. The desired trajectory is established as a combination of a linear function over time and a sinusoidal function, specifically $x_d = t$ and $y_d = \sin t$. Consequently, the target control trajectories are set as $z_{1d} = t$ and $w_{1d} = \sin t$. For the outer loop control, we utilize Eqs (19) and (21), setting the control parameters $k_1$, $l_1$, $k_2$, $l_2$ all to 10. Following the guidelines of $|v_2| \leq \alpha_2 + \beta_2 + \max \ddot{w}_d$, the control parameters $\alpha_2$ and $\beta_2$ are designed to be 3.0. Similarly, according to equation $\alpha_1 + \beta_1 \leqslant \frac{\sqrt{3}}{2}(-\alpha_2 - \beta_2 - 1 + g)$, we set $\alpha_1$ and $\beta_1$ to 1.0. In terms of inner loop control, Eq (28) guides us to select $c = 5$ and $k = 10$ to achieve a rapid convergence speed. Additionally, the differentiator is configured using Eq (29), with the parameter $\varepsilon$ set to 0.01 and the coefficients $k_1 = 9$, $k_2 = 27$, $k_3 = 27$.

## 4.3 Simulation results

The effectiveness of the control laws is clearly demonstrated by the simulation results shown in Figs 2 through 7. Fig 2 shows the performance of two control strategies in tracking a desired trajectory (x1). The upper chart displays the ideal x1 trajectory along with the tracking

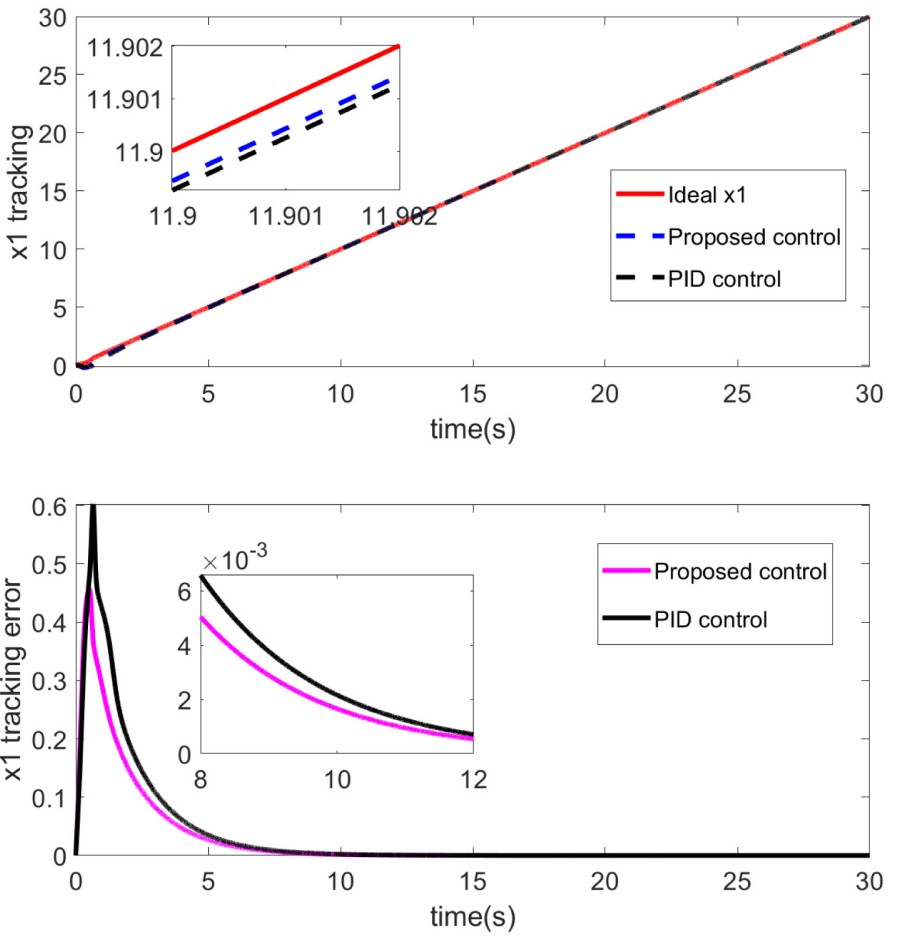

**Fig 2. The tracking and error diagram of the x1.**

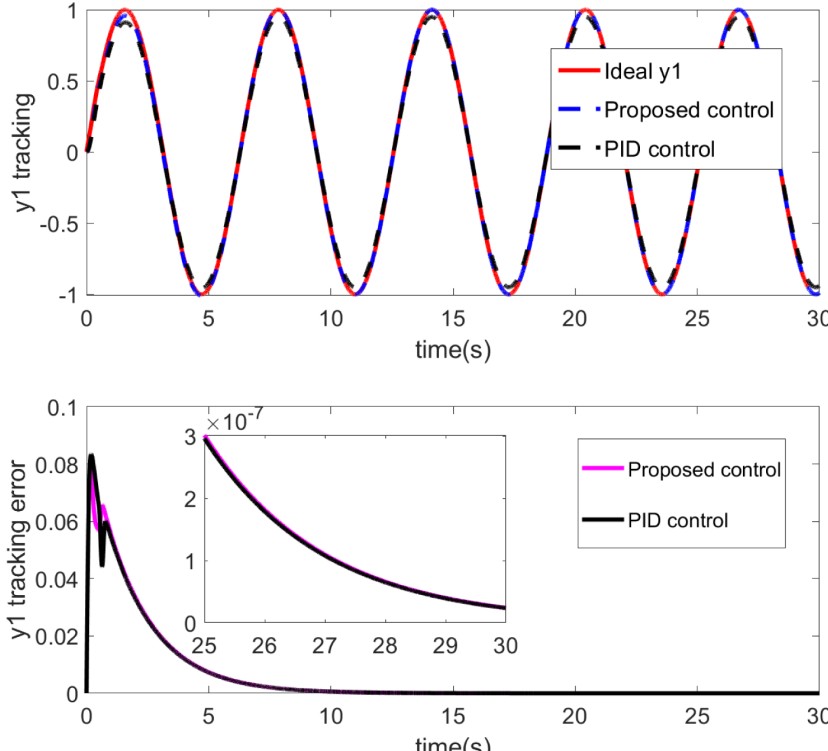

**Fig 3. The tracking and error diagram of the y1.**

performance over time of two control strategies (proposed control and PID control). The ideal trajectory is represented by a red solid line, the proposed control by a blue dashed line, and the PID control by a black solid line. It can be observed that both the proposed control strategy and the PID control strategy are effective in tracking the ideal trajectory, with the proposed control strategy being closer to the ideal trajectory for most of the time. Fig 3 illustrates the performance of two control methods-Proposed Control and PID Control-in tracking an ideal output $y1$. The red curve represents the ideal output $y1$, which is the target the system aims to achieve. The blue dashed line represents the output from the proposed control, while the black dotted line represents the output from PID control. Both control methods track the ideal output effectively, as indicated by the high degree of overlap between the curves, demonstrating small tracking errors. Overall, Fig 3 shows that the proposed control method potentially offers advantages in dynamic tracking over traditional PID control. Particularly in terms of error convergence speed and tracking precision, the proposed control appears to provide smoother and quicker performance. This may suggest that the proposed control strategy is more effective in adjusting to and responding to dynamic changes in the system. Fig 4 demonstrates the performance of two control methods—Proposed Control and PID Control—in tracking the ideal roll angle. The red curve represents the ideal roll angle (Ideal thd), which is the target the system aims to achieve. The blue dashed line indicates the output from the proposed control, and the black dotted line represents the output from PID control. The lower subplot in Fig 4 details the tracking errors from 0 seconds to 30 seconds. Fig 4 illustrates that both the proposed control and PID control exhibit good performance in the task of tracking the ideal roll angle, particularly during the stable phase after the errors rapidly decrease. Although both control

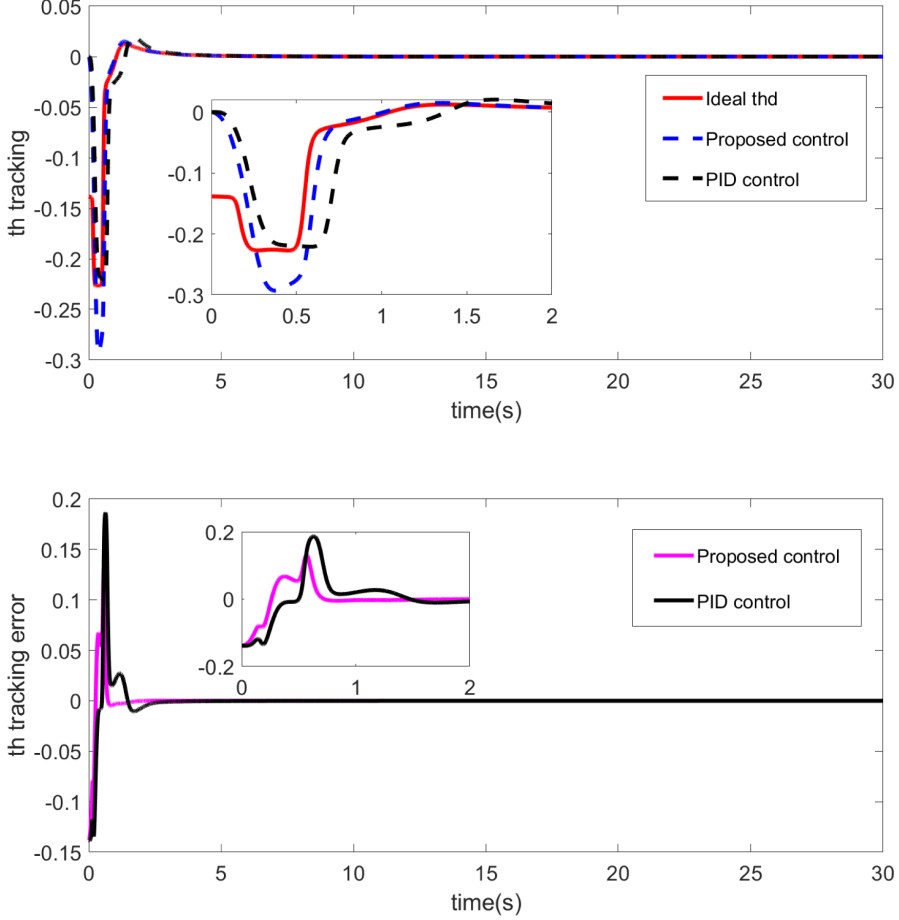

**Fig 4. Tracking of roll angle and its error response diagram.**

methods show significant errors in the initial phase, the proposed control demonstrates slightly better stability and accuracy in the later stages. This suggests that for applications requiring rapid response and high precision, the proposed control might be a more favorable choice.

Figs 5 and 6 respectively display the operational performance of two control methods-Proposed Control and PID Control-on two control inputs $u1$ and $u2$. The pink line represents the input values from the proposed control, while the black line represents the input values from PID control. In Fig 5, the control input $u1$ from both control strategies starts very high, then quickly declines, and subsequently stabilizes with periodic fluctuations. The input from the proposed control is smoother, particularly after the initial decline during the stabilization phase. In Fig 6, the input $u2$ from both control methods shows significant peaks initially, which then gradually stabilize. Between 0 and 2 seconds, both the proposed and PID control inputs exhibit intense fluctuations, after which the proposed control's input tends towards a more stable negative value, while the PID control's input remains nearly zero throughout the time frame. The proposed control method shows smoother transitions and better stability in controlling inputs $u1$ and $u2$. Especially after facing initial large fluctuations, the proposed control reaches a stable state more quickly. In summary, the proposed control method may be

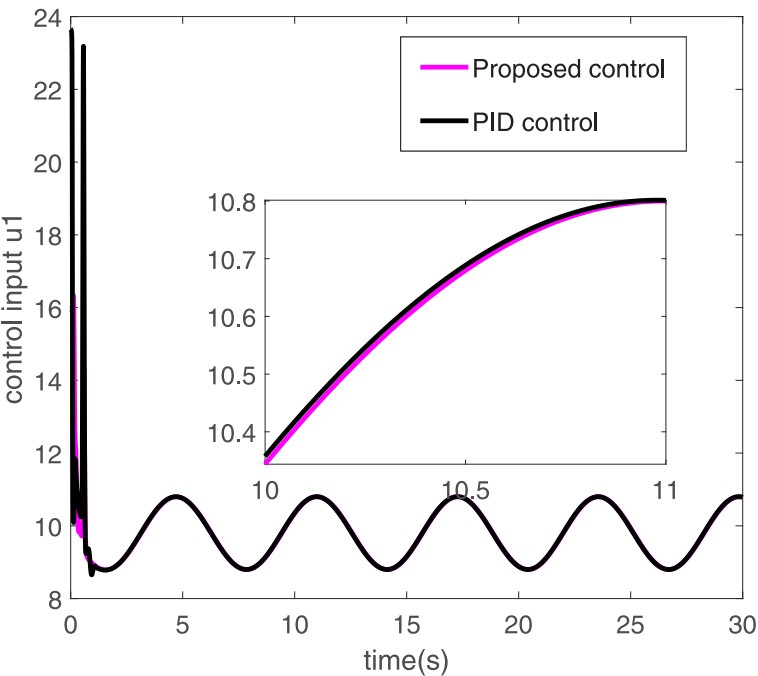

**Fig 5. Control input $u_1$.**

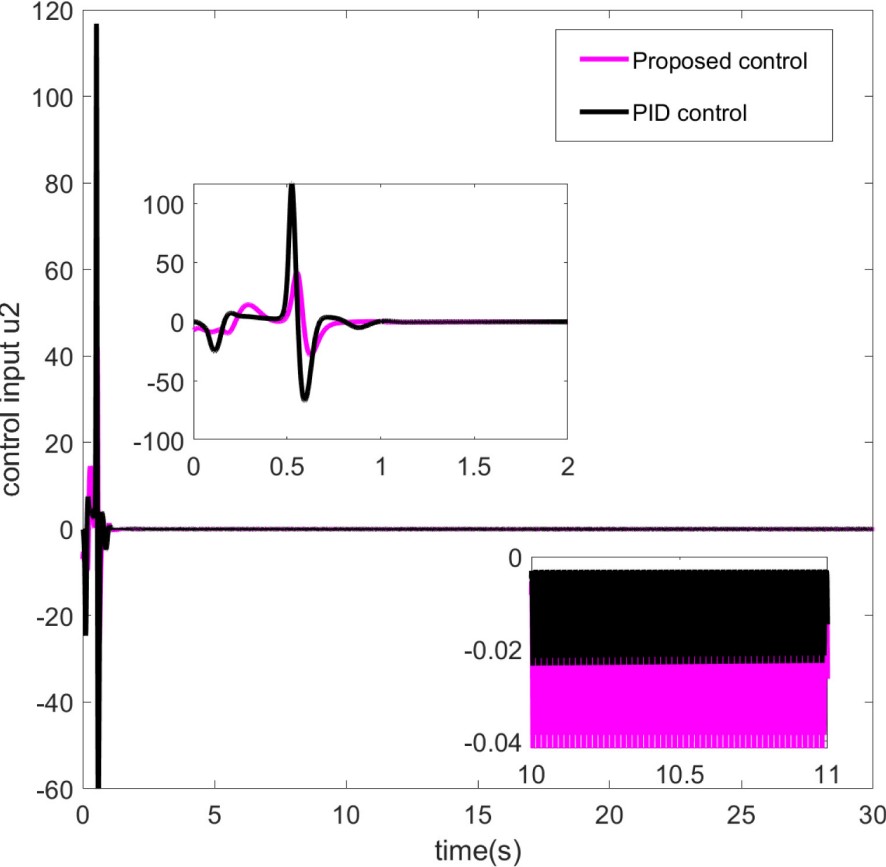

**Fig 6. Control input $u_2$.**

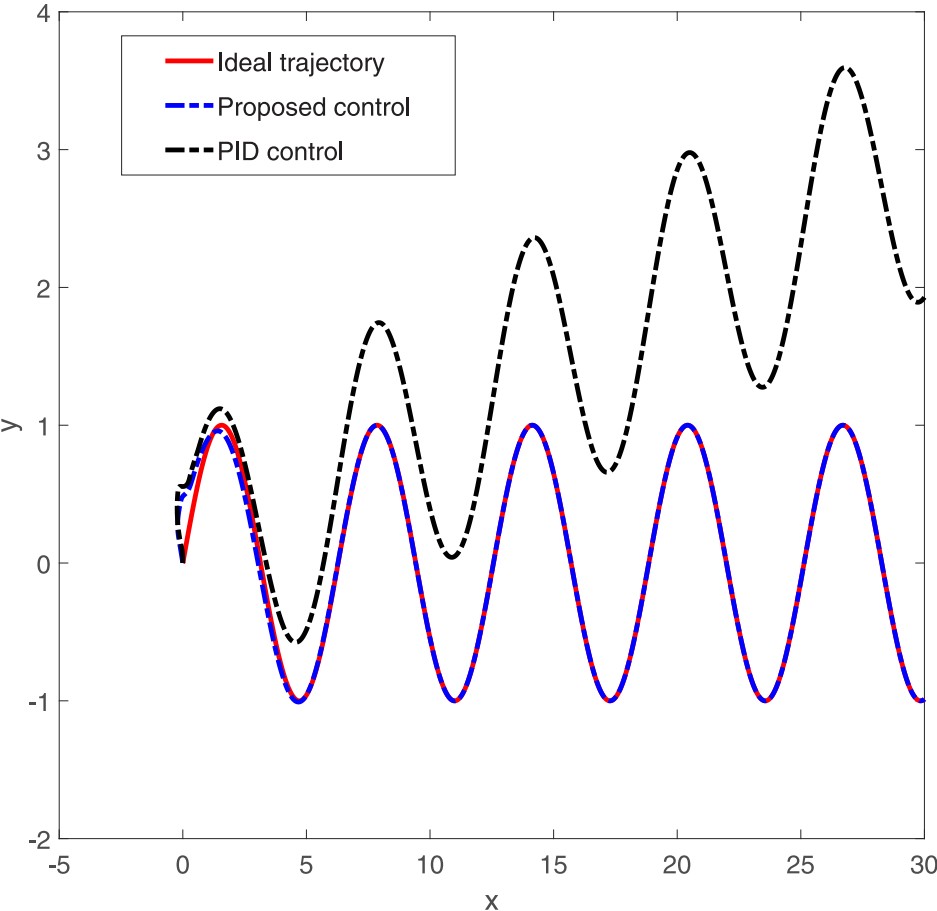

**Fig 7. Trajectory tracking response diagram.**

more effective than traditional PID control in handling rapid changes and the need for precise control in dynamic systems. Fig 7 illustrates the comparison of two control methods—Proposed Control and PID Control—in tracking a specific trajectory. The red solid line represents the ideal trajectory, which is the target path the system aims to achieve. The blue dashed line indicates the response from the proposed control, while the black dash-dotted line represents the response from PID control. From the graph, it is evident that the proposed control closely matches the ideal trajectory with very minimal deviation. Although PID control generally follows the ideal trajectory, it exhibits larger deviations in certain areas, such as peaks and troughs. Additionally, the illustration also shows that while PID control can achieve basic tracking, it may not be sufficiently precise in high-demand applications, particularly in terms of dynamic response and adjustment accuracy. Therefore, for control systems requiring high precision and stability, proposed control might be a better choice.

## 5 Discussion and future work

This study explores the flexible take-off and landing capabilities of VTOL aircraft in environments with spatial constraints, introducing a sliding mode control approach based on a double-loop architecture. This approach significantly boosts the maneuverability and reaction speed of the aircraft. The paper also discusses the application of the global asymptotic stability

theorem and designs a closed-loop system characterized by global Lipschitz continuity to ensure robust stability across both control loops. Validated through MATLAB/SIMULINK simulations, the proposed control method not only confirms its effectiveness but also increases the system's reliability and the aircraft's adaptability in dynamically complex environments. While these simulations offer an initial affirmation of the strategy's validity, the complexity of real-world scenarios necessitates further empirical testing to fully assess and refine the control method's practicality. Additionally, as VTOL aircraft find more uses in civilian and commercial sectors, enhancing their safety and reliability becomes paramount. Future research should thus prioritize evaluating the aircraft's performance under extreme conditions and investigating technological solutions to mitigate potential safety hazards.

## Supporting information

**S1 File. For the paper program, please refer to the code in supplementary materials.** (PDF)

## Author Contributions

**Conceptualization:** Liang Du.

**Data curation:** Liang Du.

**Formal analysis:** Liang Du.

**Funding acquisition:** Liang Du.

**Writing – original draft:** Liang Du.

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
