## [Decision Letter · Decision Letter 0]

10 Dec 2024

PONE-D-24-50503Trajectory tracking sliding mode control for vertical take-off and landing aircraft based on double loop and global Lipschitz stability

PLOS ONE

Dear Dr. Du,

Thank you for submitting your manuscript to PLOS ONE. After careful consideration, we feel that it has merit but does not fully meet PLOS ONE’s publication criteria as it currently stands. Therefore, we invite you to submit a revised version of the manuscript that addresses the points raised during the review process.

We look forward to receiving your revised manuscript.

Kind regards,

Omer Saleem, Ph.D.

Academic Editor

PLOS ONE

“This research is funded by the General Project of Philosophy and Social Sciences Research in Jiangsu Province Higher Education Institutions (Project Title: "Research on Entrepreneurial Models for College Students in Higher Vocational Colleges Based on 'Co-Creation between Teachers and Students," Project No. 2024SJYB0726).”

4. We note that Figure 1 in your submission contain copyrighted images. All PLOS content is published under the Creative Commons Attribution License (CC BY 4.0), which means that the manuscript, images, and Supporting Information files will be freely available online, and any third party is permitted to access, download, copy, distribute, and use these materials in any way, even commercially, with proper attribution. For more information, see our copyright guidelines: http://journals.plos.org/plosone/s/licenses-and-copyright.

Additional Editor Comments:

The paper proposed a global asymptotic stability theorem for a VTOL system and designs a closed-loop system characterized by global Lipschitz continuity, which ensures robust stability for both loops. The paper has merit and presents interesting contributions. The paper was reviewed by two separate reviewers and they both have suggested major revisions in the paper. I also believe that the paper should be majorly revised to overcome the shortcomings. The proposed revisions will improve the paper's quality.

Reviewers' comments:

Reviewer's Responses to Questions

**Comments to the Author**

1. Is the manuscript technically sound, and do the data support the conclusions?

Reviewer #1: Partly

Reviewer #2: Yes

2. Has the statistical analysis been performed appropriately and rigorously? 

Reviewer #1: No

Reviewer #2: Yes

3. Have the authors made all data underlying the findings in their manuscript fully available?

Reviewer #1: Yes

Reviewer #2: Yes

4. Is the manuscript presented in an intelligible fashion and written in standard English?

Reviewer #1: Yes

Reviewer #2: No

5. Review Comments to the Author

Reviewer #1: The results section has a scope for further improvement. Please see the comments below:

- The work is limited to a simulation environment. How the proposed SMC-based law would perform if realized on a real VTOL Aircraft?

- Include results on disturbance rejection to demonstrate the robustness of the designed SMC-based control law.

- Tracking error needs more rigorous analysis. Please do IAE, ITAE, ISE analysis.

- SMC and many of its variants suffer from the inherent phenomenon of chattering. With reference to literature such as 10.1371/journal.pone.0260480, discuss this phenomenon. How does the designed control law perform w.r.t. chattering?

- Briefly summarise the results achieved in 1-2 sentences at the end of Abstract.

- Are the Assumptions made in this study practically meaningful in a real-world scenario?

- In Section 1, start with a brief Intro on the immense need to control aerial vehicles with a particular emphasis on the highly nonlinear nature of these systems having coupled dynamics with reference to literature such as 10.3390/drones8100527. Then go to more specific case of VTOL aircraft.

- Please make sure that author's name in the literature review matches with the references. e.g. "Bacchini [6]" should be Bacchini and Cestino, "Orbea [7-9]" is not correct because not all the listed references [7-9] are from Orbear. "Wen and Chen [23]" is not correct.

- In Section 3.1 (Control Objectives), the distinguishing features of Sliding Mode Control such as robustness to disturbances and uncertainties could benefit from 10.5755/j01.eee.22.1.14094

- Add a reference under Figure 1 caption.

- In Remark 1, "R.O. Saber" needs a reference.

- 's' or 'seconds'? Please make it consistent.

Reviewer #2: As a reviewer, I have carefully examined the paper titled "Trajectory tracking sliding mode control for vertical take-off and landing aircraft based on double loop and global Lipschitz stability". It presents a sliding mode control (SMC) strategy for VTOL aircraft using a dual-loop system and global Lipschitz stability. This approach addresses significant challenges in VTOL control. But there are few limitations that could potentially be improved:

1. Need to improve the abstract, i.e., address the problem clearly, and impact of your research

2. Why you used SMC? why you didn't use PID, MPC or other methods? it lacks a comprehensive comparison with other state-of-the-art control strategies.

3. In abstract you used some words which need to be change like corroborated etc.

4. You mentioned that the VTOL are agile or agility but it is not agile as fixed wing UAS, so please change it.

5. some time you used VTOL aircraft, somewhere you used VTOL drone, please stick on one word, drone or aircraft.

6. While the introduction mentions the importance of VTOL aircraft, it doesn't provide enough specific context about current challenges in VTOL control systems.

7. All or most of the UAS have dual-loop, inner and outer: inner for attitude and outer loop for positions and altitude. As the speed or frequency of inner loop is higher than outer loop. So what is your contribution here?

8. The specific problem that this research aims to address is not clearly articulated in the introduction.

9. Clear research questions or objectives are not explicitly stated in the introduction.

10. Please make a Remark 1 and Remark 2 more concise and clear.

11. Please cite the R.O. Saber in line 96

12. In eq. 1, the second line should be y_ddot not y_dot (should be double dot)

13. Please explain how you get the two decoupling eqs. in eq. 3,

14. In assumption 1: Why you consider it as a linear system, your equation is totally nonlinear

15. Please explain mathematically two lines on how you define the lyapunov function?

16. In line 183 - 185 the desired reference should be in super_subscript, it shows you missed the underline in latex

17. In line 186 and line 190, you means Lemma 1 not theorem 1, because there is no theorem you defined.

18. Please define one Lemma about what is Lipschitz, and how eq. 15 and eq. 17 ensured globally Lipschitz continuous

19. Please define shortly what is SMC law in line 217

20. After remark 4, please start line 224 from new line, as In control rule ......

21. How you select the x3_dot and epsilon in eq 26

22. In line 239, the equation number should be 15 and 17, please correct it.

23. It seems that the figures are stretched, please use the proper and default size of figures according to the requirements of the journal.

24. Please adjust all the simulation figures in one or two pages, your one figure i.e., Fig 8 in one page, please adjust the figures according to the journal requirements.

25. Need comparison with other state-of-the-art control strategies in your simulation results.

26. Your simulation code is not working, I tested on my system with MATLAB version 2024a.

27. In you didn't try it on hardware than please try the 3D simulation in MATLAB.

28. While the paper addresses stability, there is limited discussion on the robustness of the control system to various disturbances or uncertainties that might be encountered in real-world operations.

29. The proposed sliding mode control strategy, while theoretically sound, appears to be quite complex. This complexity might pose challenges in real-world implementation and tuning of the control system. How you justify this question?

30. Please add one Figure, in which it will show the schematic diagram, in which show the flow of control to actuators of VTOL, remove the Fig. 1 and add one complete schematic diagram in which you put the vtol in the place of system or plant.

31. English is very weak, Need to much improvements.

32. Grammer and punctuations also need to be improved.

Thank you

6. PLOS authors have the option to publish the peer review history of their article (what does this mean?). If published, this will include your full peer review and any attached files.

Reviewer #1: No

Reviewer #2: **Yes: **Muhammmad kazim

---

## [Author Response · Author response to Decision Letter 0]

22 Dec 2024

Please see the file uploaded by the system.

---

## [Decision Letter · Decision Letter 1]

7 Jan 2025

PONE-D-24-50503R1Trajectory tracking sliding mode control for vertical take-off and landing aircraft based on double loop and global Lipschitz stabilityPLOS ONE

Dear Dr. Du,

Thank you for submitting your manuscript to PLOS ONE. After careful consideration, we feel that it has merit but does not fully meet PLOS ONE’s publication criteria as it currently stands. Therefore, we invite you to submit a revised version of the manuscript that addresses the points raised during the review process.

We look forward to receiving your revised manuscript.

Kind regards,

Gang Wang

Academic Editor

PLOS ONE

Journal Requirements:

Reviewers' comments:

Reviewer's Responses to Questions

**Comments to the Author**

1. If the authors have adequately addressed your comments raised in a previous round of review and you feel that this manuscript is now acceptable for publication, you may indicate that here to bypass the “Comments to the Author” section, enter your conflict of interest statement in the “Confidential to Editor” section, and submit your "Accept" recommendation.

Reviewer #1: (No Response)

Reviewer #2: All comments have been addressed

2. Is the manuscript technically sound, and do the data support the conclusions?

Reviewer #1: Yes

Reviewer #2: Yes

3. Has the statistical analysis been performed appropriately and rigorously? 

Reviewer #1: I Don't Know

Reviewer #2: Yes

4. Have the authors made all data underlying the findings in their manuscript fully available?

Reviewer #1: Yes

Reviewer #2: Yes

5. Is the manuscript presented in an intelligible fashion and written in standard English?

Reviewer #1: Yes

Reviewer #2: Yes

6. Review Comments to the Author

Reviewer #1: Authors have addressed all the suggested changes. The revised version of the paper has been significantly improved. The paper can now be accepted.

Reviewer #2: Thank you for your revision and clarification of most comments. However, there are still areas for improvement:

1. Please add a figure showing the schematic diagram and control flow to VTOL actuators, as requested in the first revision.

2. Provide a comparison of the proposed controller with other controllers like PID, MPC, or higher-order sliding modes controllers.

3. Reduce gaps between figures by placing four figures on one page.

4. Include simulations of:

a) Vertical takeoff, cruise, and landing

b) Takeoff, circular motion, and landing

Show errors with reference track for both scenarios.

5. Further improvements needed in the simulation section.

7. PLOS authors have the option to publish the peer review history of their article (what does this mean?). If published, this will include your full peer review and any attached files.

Reviewer #1: No

Reviewer #2: No

---

## [Author Response · Author response to Decision Letter 1]

10 Jan 2025

Please see the upload attachment.

---

## [Editor Report · Decision Letter 2]

14 Jan 2025

Trajectory tracking sliding mode control for vertical take-off and landing aircraft based on double loop and global Lipschitz stability

PONE-D-24-50503R2

Dear Dr. Du,

We’re pleased to inform you that your manuscript has been judged scientifically suitable for publication and will be formally accepted for publication once it meets all outstanding technical requirements.

Kind regards,

Gang Wang

Academic Editor

PLOS ONE
---

## [Editor Report · Acceptance letter]

23 Jan 2025

PONE-D-24-50503R2 

PLOS ONE

Dear Dr. Du, 

I'm pleased to inform you that your manuscript has been deemed suitable for publication in PLOS ONE. Congratulations! Your manuscript is now being handed over to our production team.

Kind regards, 

on behalf of

Dr. Gang Wang 

Academic Editor

PLOS ONE